# Microsaccades as a marker not a cause for attention-related modulation

**Gongchen Yu**[1]*, **James P Herman**[2], **Leor N Katz**[1], **Richard J Krauzlis**[1]*

[1]Laboratory of Sensorimotor Research, National Eye Institute, Bethesda, United States; [2]Department of Ophthalmology, University of Pittsburgh School of Medicine, Pittsburgh, United States

**Abstract** Recent evidence suggests that microsaccades are causally linked to the attention-related modulation of neurons—specifically, that microsaccades toward the attended location are required for the subsequent changes in firing rate. These findings have raised questions about whether attention-related modulation is due to different states of attention as traditionally assumed or might instead be a secondary effect of microsaccades. Here, in two rhesus macaques, we tested the relationship between microsaccades and attention-related modulation in the superior colliculus (SC), a brain structure crucial for allocating attention. We found that attention-related modulation emerged even in the absence of microsaccades, was already present prior to microsaccades toward the cued stimulus, and persisted through the suppression of activity that accompanied all microsaccades. Nonetheless, consistent with previous findings, we also found significant attention-related modulation when microsaccades were directed toward, rather than away from, the cued location. Thus, despite the clear links between microsaccades and attention, microsaccades are not necessary for attention-related modulation, at least not in the SC. They do, however, provide an additional marker for the state of attention, especially at times when attention is shifting from one location to another.

**\*For correspondence:**
yugongchen1990@gmail.com
(GY);
richard.krauzlis@nih.gov (RJK)

**Competing interest:** The authors declare that no competing interests exist.

## Editor's evaluation

This is very much needed work, especially in light of the recent debate regarding whether or not microsaccades are the cause of peripheral attentional effects. A few influential papers have been published recently strongly suggesting that attentional effects are primarily the result of the execution of tiny microsaccades that humans/primates perform during fixation while attending to peripheral stimuli. These past findings have, naturally, a number of implications for the way we interpret visual attention, and raised the question of whether shifts of attention are dependent on microsaccades. By explicitly comparing and quantifying the effects of attention on neuronal responses in the presence and in the absence of microsaccades, this work provides important insights into this debate.

## Introduction

The allocation of visual spatial attention is associated with both the enhancement of neuronal activity at the attended location (**Desimone and Duncan, 1995**; **Krauzlis et al., 2013**; **Reynolds and Chelazzi, 2004**) and the tendency to make microsaccades toward the covertly attended stimulus while fixating (**Engbert and Kliegl, 2003**; **Hafed and Clark, 2002**; **Lowet et al., 2018**). A recent study provided evidence that the generation of microsaccades could play a causal role in the attention-related modulation of neuronal activity (**Lowet et al., 2018**). In a spatial attention task in which subjects were

rewarded for making a saccade to a cued stimulus after it changed color, cortical neurons displayed attention-related enhancement only following microsaccades directed toward the attended stimulus.

These findings provide interesting evidence about the links between covert attention, neuronal modulation, and fixational eye movements. However, they also raise potentially serious questions about how to interpret neuronal data obtained during the fixation tasks commonly used to study visual attention. Given that microsaccades are unavoidably generated during fixation, could the well-known modulation of neurons during covert visual attention tasks be an artifact of microsaccade generation? Is attention-related neuronal modulation obligately tied to the oculomotor intention to orient toward the attended stimulus, as suggested by the premotor theory of attention (*Rizzolatti et al., 1987*), and this dependence has been overlooked because sometimes it is a microsaccade toward the attended stimulus?

Here, we tested the relationship between microsaccade generation and attention-related neuronal modulation in the primate superior colliculus (SC), one of the most important brain structures for the control of visual spatial attention. Neurons in the SC provided the first evidence for neural correlates of visual attention (*Goldberg and Wurtz, 1972*) and are modulated during both overt and covert attention tasks (*Herman and Krauzlis, 2017*; *Ignashchenkova et al., 2004*). Most relevant to our questions, SC activity is causally related to behavioral performance in attention tasks: inactivation causes attention deficits specifically for the affected location and conversely, microstimulation can selectively facilitate attention performance (*Cavanaugh and Wurtz, 2004*; *Herman et al., 2018*; *Lovejoy and Krauzlis, 2010*; *Müller et al., 2005*). Thus, if microsaccades are a necessary part of allocating attention, their causal role should be evident in the activity of SC neurons.

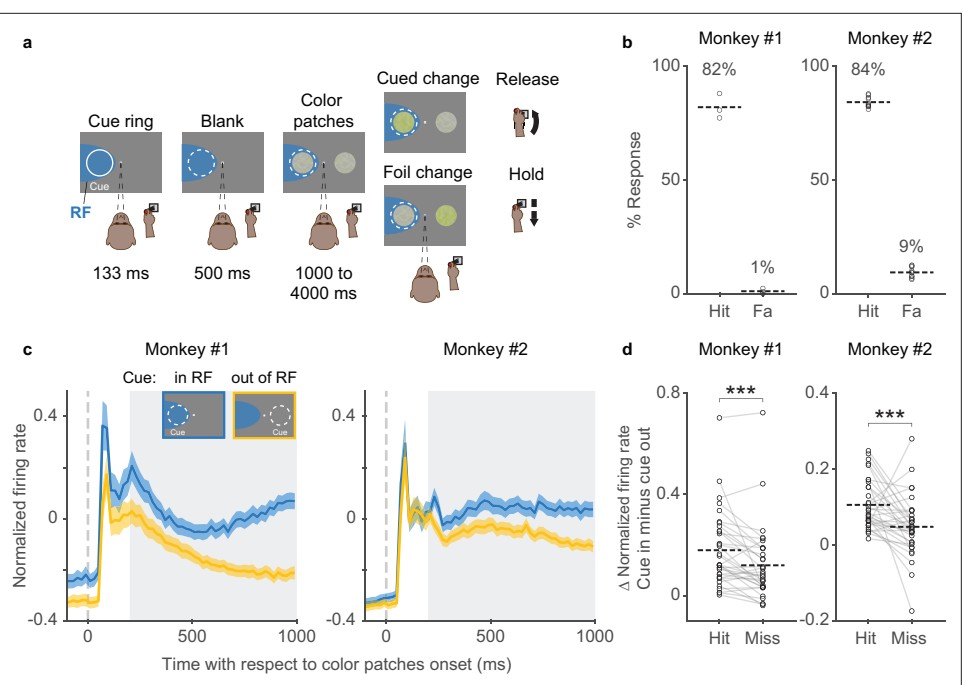

**Figure 1.** Behavioral performance and SC neuronal activity in a covert spatial attention task. (**a**) The monkey was required to maintain central fixation, releasing a joystick in response to a color change at the cued location and holding their response if the change occurred at the opposing foil location. The dashed white ring illustrates the cued location and the blue shaded area denotes the response field (RF) of SC neurons; neither were visible to the monkey. (**b**) Hit rates (Hit) and false-alarm rates (Fa) for monkeys 1 and 2 in each session. Each circle represents data from one behavioral session. Percentages and horizontal dashed lines denote average hit rates and false-alarm rates across sessions. (**c**) Population SC average normalized firing rates for cue-in-RF (blue) and cue-out-of-RF (yellow) conditions, aligned on the onset of the color patches. The insets illustrate the cue conditions when the SC RFs were on the left side. The gray shaded areas denote the time windows (the delay period) used for measuring the difference (Δ) in average normalized firing rates for (**d**). The difference (Δ) in average normalized firing rates between cue-in-RF and cue-out-of-RF of SC neurons in hit and miss trials. Each pair of circles connected by a gray line represents the data from one SC neuron. Horizontal dashed lines denote the averages across neurons. *** denotes p<0.001, Wilcoxon signed-rank test. SC, superior colliculus.

# Results

To investigate the possible role of microsaccades in attention-related modulation in the SC, we collected neuronal activity and eye movement data in two monkeys (*Macaca mulatta*) during a covert spatial attention task that has been used in our previous studies examining the neuronal mechanisms of attention (*Herman et al., 2018*; *Herman and Krauzlis, 2017*). SC extracellular activity was recorded with linear electrode arrays and eye position was measured with an infrared eye-tracker.

In the covert spatial attention task (*Figure 1a*), head-fixed monkeys were required to release a joystick in response to a change in color saturation for the stimulus patch at the cued location and to withhold their response if the change occurred at an opposing foil location. Monkeys were required to maintain central fixation throughout the entire trial such that any allocation of attention to the cued stimulus patch was only done covertly. The behavioral performance was very consistent across sessions (*Figure 1b*). Monkey subjects had hit rates of ~80% (mean hit rates: 82% in monkey 1% and 84% in monkey 2), as expected since the size of the change in color saturation was set near the monkeys' detection thresholds as described previously (*Herman and Krauzlis, 2017*), and the monkeys rarely responded to the foil changes (mean false-alarm rates: 1% in monkey 1 and 9% in monkey 2), indicating that they followed the attention cue and attended to the correct stimulus location.

While monkeys performed the task, many SC neurons displayed attention-related modulation. The time course of SC population firing rate with respect to the onset of the color patches onset is shown in *Figure 1c* for both monkeys (34 units in monkey 1 and 34 units in monkey 2). SC neurons displayed a phasic response shortly after the onset of the color patches, and during the delay period (200–1000 ms) that followed, neuronal activity became steadily higher when the cued stimulus was inside the response field (RF) of the SC neurons compared to when it was outside the RF. This pattern recapitulates the well-known pattern of attention-related modulation in the SC and elsewhere (*Desimone and Duncan, 1995*; *Herman et al., 2018*; *Herman and Krauzlis, 2017*; *Krauzlis et al., 2013*; *Reynolds and Chelazzi, 2004*).

To confirm that SC attention-related modulation was indeed linked to performance in our attention task, we compared the modulation on correctly and incorrectly performed trials (*Figure 1d*). We measured the cue-related difference in firing rate in each neuron (i.e., cue-in-RF minus cue-out-of-RF) during the delay period (gray shaded

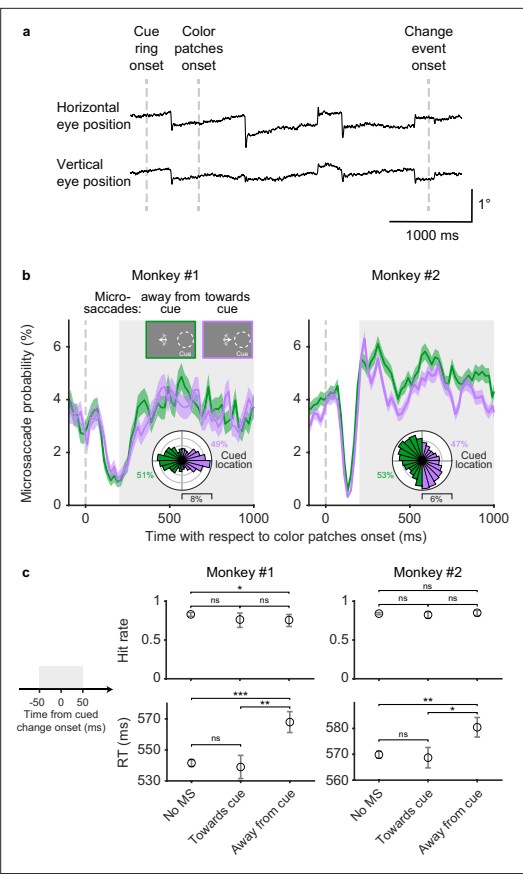

**Figure 2.** Microsaccades in the covert attention task. (**a**) Single-trial horizontal and vertical eye position traces. The abrupt deflections are microsaccades. (**b**) Session average probability of microsaccades toward the cued location (purple) and away from the cued location (green), aligned on color patches onset. The insets at top illustrated the microsaccade conditions when the cue was on the right side. The white arrows schematically represent microsaccades. The polar plots show the directional distribution of microsaccades during the delay period (gray shaded area), relative to the cued location. The numbers beside the polar plots denote the total proportion of microsaccades toward and away from the cued location. Error bars denote the standard error of the mean (SEM). (**c**) Average hit rates (top row) and reaction times (RT, bottom row) when there were no microsaccades, microsaccades toward the cued location, and microsaccades away from the cued location –50 to 50 ms relative to cued change onset (left schematic, gray shaded area). Error bars of hit rates denote the 95% binomial confidence interval. Error bars of reaction times denote the SEM. *** denotes $p < 0.001$, ** denotes $p < 0.01$, * denotes $p < 0.05$, 'ns' denotes $p > 0.05$, chi-square proportion test for hit rates and Wilcoxon rank-sum test for reaction times.

area in *Figure 1c*) separately for hit (correct detections) and miss trials. We found that SC neurons displayed significantly higher average attention-related modulation in hit trials compared to miss trials (monkey 1: p=0.0004, monkey 2: p=0.00099, Wilcoxon signed-rank test), verifying the link between SC neuronal activity and performance in this attention task (*Herman et al., 2018*).

Simultaneous to SC neuronal recording, we measured the subjects' fixational eye movements. Traces of horizontal and vertical eye position from one example trial are shown in *Figure 2a*. Even though monkeys maintained fixation throughout the trial, they generated a series of fixational microsaccades consisting of abrupt and miniscule (typically smaller than 1°) deflections in eye position, as expected from previous findings (*Hafed, 2011*; *Martinez-Conde et al., 2013*; *Rucci and Poletti, 2015*).

To summarize when microsaccades occurred and where they were directed during the covert attention task, we calculated the probability of microsaccades on the same time axis as the SC neuronal firing rate (*Figure 2b*). To identify how the direction of microsaccades was influenced by the location of the spatial cue, microsaccades were grouped based on whether they were directed toward or away from the cued hemifield. Overall, the probability of microsaccades both toward and away from the cued hemifield decreased sharply after patch onset and then rebounded during the following delay period. During the delay period (*Figure 2b*, gray shaded area), the probability of microsaccades toward and away from the cued hemifield were similar, with slightly higher probability of away microsaccades in monkey 2. The polar plots in *Figure 2b* provide a more detailed picture of the directional distributions of microsaccades relative to the cued location during the delay period. Microsaccades in monkey 1 showed two predominant directions directly toward and away from the cued location, whereas microsaccades in monkey 2 showed a weaker cue-related bias and idiosyncratic vertical tendencies unrelated to the cue. Thus, during sustained attention in the delay period, monkey

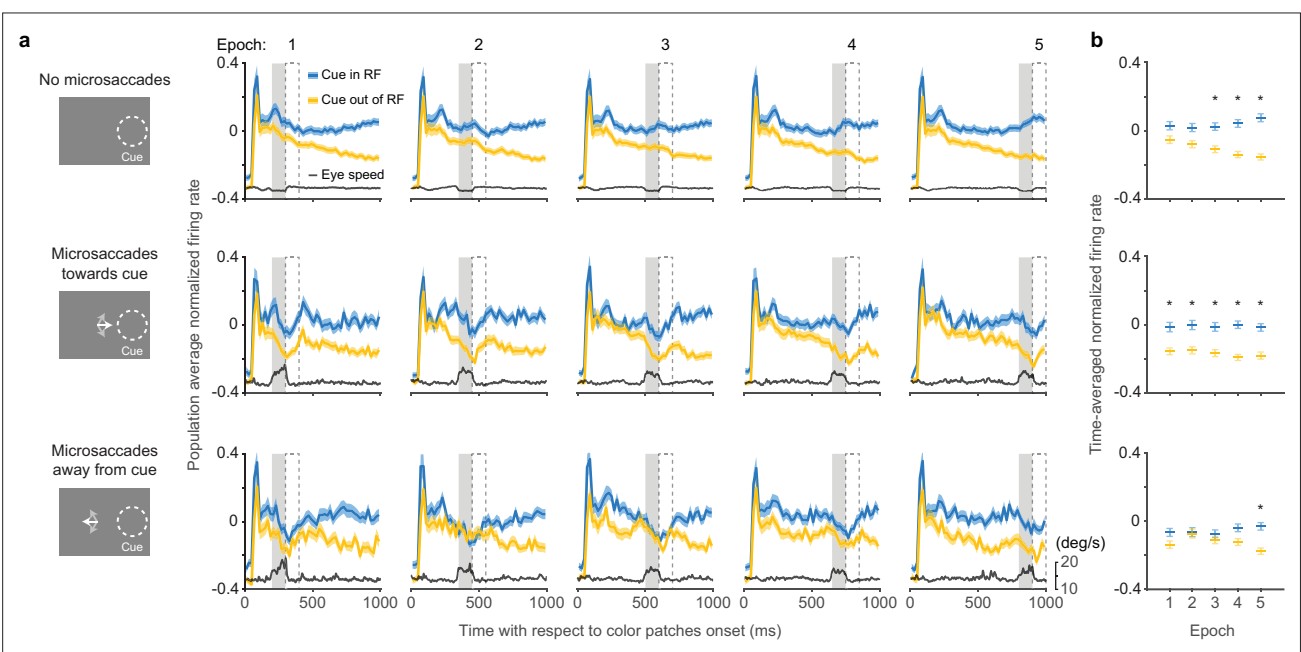

**Figure 3.** Effects of microsaccades on the time course of SC attention-related modulation. (**a**) Each panel depicts population average SC normalized firing rates (blue, cue-in-RF; yellow, cue-out-of-RF) and eye speed traces (black) from subsets of trials in which there were either no microsaccades (top row), microsaccades toward the cue (middle row), or microsaccades away from the cue (bottom row), within a particular 100-ms epoch indicated by the gray shaded area. The chosen time epochs were: 200–300 ms, 350–450 ms, 500–600 ms, 650–750 ms, and 800–900 ms after color patches onset. The 100-ms dashed boxes following the gray shaded areas denote the time windows used for measuring time-averaged normalized firing rates for (**b** ). Time-averaged normalized firing rates as a function of epoch. The asterisks denote the epochs with significant higher activity for cue-in-RF than for cue-out-of-RF, p<0.05, ANOVA, Tukey-Kramer post hoc comparisons. Error bars denote SEM. SC, superior colliculus.

The online version of this article includes the following figure supplement(s) for figure 3:

**Figure supplement 1.** Effects of microsaccades on the time course of SC attention-related modulation were similar when using other measurement windows.

subjects generated frequent microsaccades but their overall direction was not systematically biased toward the cued location.

Although microsaccades during the delay period were not biased toward the cue, microsaccade direction was associated with some differences in task performance (*Figure 2c*). These effects were identified by subdividing the monkeys' hit rates (*Figure 2c*, first row) and reaction times (*Figure 2c*, second row) into three sets based on the microsaccades in a 100-ms window centered on the change onset: no microsaccades, microsaccades toward the cued location, and microsaccades away from the cued location. We observed only minor differences in hit rate but consistent effects on reaction times. Both monkeys had slower reaction times when microsaccades were directed away from the cued location compared to when there were no microsaccades (monkey 1: p=0.0002, monkey 2: p=0.002, Wilcoxon rank-sum test) or microsaccades toward the cued location (monkey 1: p=0.002, monkey 2: p=0.016, Wilcoxon rank-sum test).

## Effects of microsaccades on the time course of SC attention-related modulation

We next assessed how the occurrence of microsaccades was related to the time course of SC attention-related modulation (*Figure 3*). Each panel in *Figure 3a* depicts the population average normalized SC firing rate from subsets of trials in which there were either no microsaccades (top row), microsaccades toward the cued location (middle row), or microsaccades away from the cued location (bottom row), within a particular 100-ms epoch indicated by the gray shaded area. The eye speed trace in each panel serves to verify the inclusion or exclusion of microsaccades within each epoch. In total, we used five different epochs during the delay period, corresponding to the columns in *Figure 3a*. To summarize the neuronal data, we measured the average normalized firing rates in 100-ms windows (dashed boxes) immediately after each microsaccade epoch and plotted these averages as a function of epoch in *Figure 3b*. This choice of measurement window was motivated by the recent finding that cortical attention-related modulation emerged ~100 ms after microsaccade onset (*Lowet et al., 2018*), but we found similar results with other windows (*Figure 3—figure supplement 1*).

In trials with no microsaccades in each epoch, the pattern of attention-related neuronal modulation (*Figure 3a*, top row) was qualitatively similar to the one observed when including all trials (*Figure 1c*). We found steadily higher population average neuronal activity during 'cue-in-RF' condition compared to 'cue-out-of-RF condition' and this attention-related modulation was significant from the third epoch onwards (*Figure 3b*, top row, ANOVA, post hoc comparisons, p<0.05). Thus, the exclusion of trials with microsaccades did not prevent the emergence of attention-related modulation over time during the delay period.

In trials with microsaccades in each epoch, SC neuronal activity was suppressed. Distinct dips in the firing rate traces were evident toward the end of each epoch, regardless of whether the microsaccades were directed toward the cue (*Figure 3a*, middle row), or away from the cue (*Figure 3a*, bottom row).

In addition to this suppression of firing rate that occurred regardless of microsaccade direction, we also found an attention-related modulation that did vary with the direction of the microsaccades. In trials with microsaccades toward the cued location (*Figure 3a*, middle row), the difference between neuronal activity for 'cue-in-RF' versus 'cue-out-of-RF' conditions was significant in all the epochs (*Figure 3b*, middle row, ANOVA, post hoc comparisons, all p<0.05). In contrast, for trials with microsaccades away from the cued location (*Figure 3a*, bottom row), the attention-related modulation was reduced and only emerged as significant in the last time epoch (*Figure 3b*, bottom row, ANOVA, post hoc comparison, p<0.05).

Thus, the presence of SC attention-related modulation did not require the generation of microsaccades. Instead, microsaccades were associated with the suppression of firing rates, consistent with saccadic suppression (*Hafed and Krauzlis, 2010*). Nonetheless, there was a systematic relationship between the amplitude of the attention-related modulation and the direction of the microsaccades—we found more consistent attention-related modulation when microsaccades were directed toward the cue but not when they were directed away, suggesting that there might be a causal relationship between the two.

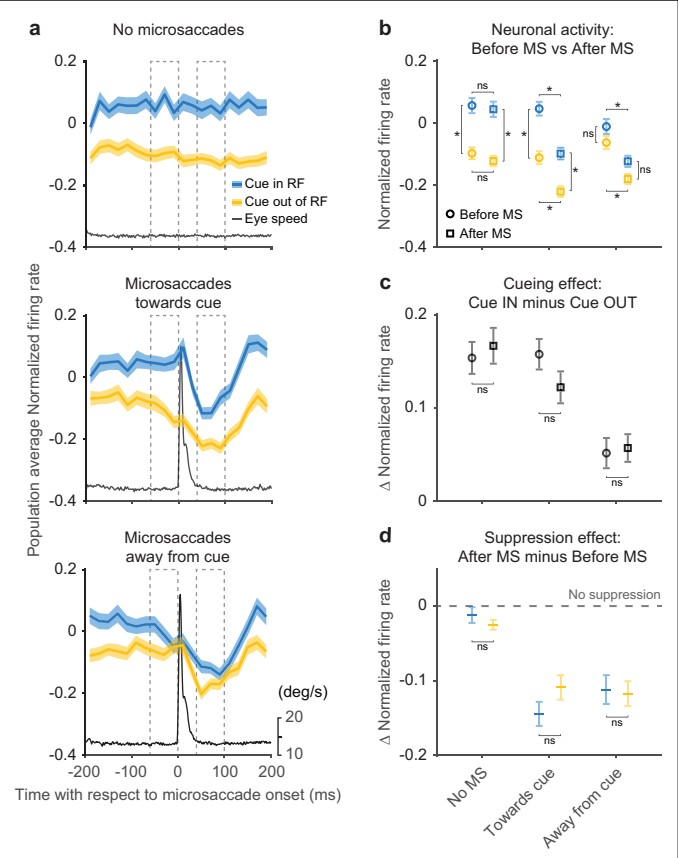

**Figure 4.** Peri-microsaccadic attention-related modulation. (**a**) Population average SC normalized firing rates aligned to the onset of individual microsaccades under three conditions: timing-matched no microsaccades (top), microsaccades toward cue (middle), and microsaccades away from cue (bottom). The gray line denotes the average eye speed. The dashed boxes depict the windows we used to calculate the average normalized firing rates before (−60 to 0 ms) and after (40–100 ms) microsaccades. (**b**) Average normalized firing rates before (circle) and after (square) microsaccades. (**c**) The difference (Δ) in average normalized firing rates between cue-in-RF and cue-out-of-RF during 'before microsaccade' (circle) and 'after microsaccade' (square) windows. (**d**) The difference (Δ) in average normalized firing rates between 'after' and 'before' window for cue-in-RF (blue) and cue-out-of-RF (yellow) conditions. The dashed line indicates the level of 'no suppression.' The asterisk denotes p<0.05 and 'ns' denotes p>0.05, ANOVA, Tukey-Kramer post hoc comparisons. Error bars denote SEM. SC, superior colliculus.

The online version of this article includes the following figure supplement(s) for figure 4:

**Figure supplement 1.** Peri-microsaccadic attention-related modulation was similar when using narrower windows to group microsaccades toward/away from the cued location.

**Figure supplement 2.** Peri-microsaccadic attention-related modulation was robust to the thresholds for microsaccade detection.

**Figure supplement 3.** Peri-microsaccadic attention-related modulation was not explained by motor effects or differences in eye position.

## Peri-microsaccade attention-related modulation

We next investigated the timing of the attention-related modulation relative to microsaccades by aligning SC neuronal activity to the onset of individual microsaccades directed toward (*Figure 4a*, middle row) or away from the cued location (*Figure 4a*, bottom row). For comparison, we also generated control data sets that contained no microsaccades during 400-ms windows chosen to match the timing of our microsaccade-aligned data (*Figure 4a*, top row). In these 'no microsaccades' data, we still observed substantial attention-related modulation, again confirming that microsaccades were not necessary for SC attention-related modulation.

For the data aligned on microsaccades, neuronal activity was suppressed immediately after microsaccade onset regardless of microsaccade direction or cue conditions (*Figure 4a*), consistent with the results in *Figure 3*. In contrast, the amplitude of the attention-related modulation did vary with microsaccade direction—the modulation was present only when the microsaccade was directed toward rather than away from the cued location. These results were robust with respect to the particular microsaccade inclusion window used for the analysis (*Figure 4—figure supplement 1*) and also to the choice of threshold used to detect microsaccades (*Figure 4—figure supplement 2*).

The difference in attention-related modulation between microsaccade conditions (toward vs. away from cue) was not triggered by the microsaccade but was evident before microsaccade onset. To quantify changes in firing rates and attention-related modulation around the time of microsaccades, we calculated the average normalized firing rate before and after microsaccade onset (*Figure 4b*). The dashed boxes in *Figure 4a* depict the 'before' and 'after' windows we used for the calculation. For the 'no microsaccades' and 'toward cue' data sets, we found significant cue-related modulation (firing rate with cue-in-RF was higher than with cue-out-of-RF) in both the 'before' and 'after' windows (ANOVA, post hoc comparisons, all p<0.05). In contrast, for the 'away from cue' data sets, we did not find significant attention-related modulation in either the 'before' or 'after' window (ANOVA, post hoc comparisons, all p>0.05). We also found a significant suppression of neuronal activity in the 'after' window compared to 'before,' regardless of cue condition, for both microsaccade directions (ANOVA, pos hoc comparisons, all p<0.05), but not for the no-microsaccade condition (p>0.05).

As there were both attention-related modulation and response suppression around the occurrence of microsaccades, we investigated how these two processes interacted with each other. First, we tested whether the attention-related modulation present before microsaccades was changed when the overall neuronal response was suppressed after microsaccades (*Figure 4c*). To compare the attention-related modulation, we measured the difference in population average normalized firing rates between cue-in-RF and cue-out-of-RF conditions during both 'before' and 'after' windows. We did not observe any significant differences in attention-related modulation between 'before' and 'after' windows for any of the three microsaccade-related data sets (ANOVA, post hoc comparisons, all p>0.05). Thus, the amplitude of the attention-related modulation was largely preserved through the occurrence of the microsaccade.

Next, we investigated whether the microsaccade-related neuronal suppression observed in *Figure 4b* was affected by the cueing condition (*Figure 4d*). We calculated the difference in normalized firing rates between 'after' and 'before' windows for both 'cue-in-RF' and 'cue-out-of-RF' conditions. We did not find any significant differences in the amplitude of the suppression effect across cueing conditions for any of the three microsaccade-related conditions (ANOVA, post hoc comparisons, all p>0.05). This indicates that the amplitude of the microsaccade-related suppression was not influenced by the cue-related modulation. Finally, we confirmed that our microsaccade-related findings could not be explained by motor effects related to the generation of microsaccades or variations in eye position (*Figure 4—figure supplement 3*).

In summary, aligning firing rates on microsaccade onset clarifies the causal relationship between attention-related modulation and microsaccades. SC neurons display large attention-related modulation during epochs that contain no microsaccades. When microsaccades do occur, the attention-related modulation is present before microsaccade onset and has an amplitude at least as large as that found after microsaccades. Thus, microsaccades do not appear to cause the attention-related modulation but may be influenced by a shared process.

## Discussion

Our findings demonstrate that microsaccades are not necessary for attention-related modulation in the midbrain SC. First, attention-related modulation emerged over the same time course during the attention task, regardless of the occurrence of microsaccades (*Figure 3*). Second, attention-related modulation was still observed when microsaccades were completely absent over long time periods (*Figure 4*). Third, when microsaccades did occur, the neuronal attention-related modulation was already present prior to microsaccade onset and the occurrence of the microsaccade—and the accompanying saccade-related suppression—did not flip the pre-existing trend of attention-related modulation (*Figure 4*). Thus, the attention-related modulation of SC neurons was readily

dissociated from the occurrence of microsaccades during the sustained allocation of visual spatial attention.

Why are our findings different from those in a recent study showing that microsaccades might play a causal role in the neuronal attention-related modulation (*Lowet et al., 2018*)? There are several factors that could have contributed to this difference. The first factor is the time window used for microsaccade analysis. The previous study focused on microsaccades immediately after cue onset. Thus, the cue-induced change in attention state might be expected to trigger changes in both fixational eye movements and neuronal modulation. In our study, the time window we focused on, the 'delay period,' is hundreds of milliseconds after the cue, and is the time window used in most neurophysiological studies of attention (*Bogadhi et al., 2021*; *McAdams and Maunsell, 1999*). The second factor is how the spatial cues were presented. In our paradigm, the cue ring briefly appeared in the periphery and then disappeared. In contrast, the previous paradigm used a cue presented near fixation that persisted throughout the trial. Our cue therefore provided less of an impetus to generate small saccades directed toward the cue, compared to the case when the cue is continuously near the center of gaze. The third factor relates to response modality. In our task, monkeys were trained to release a joystick to report their detection of stimulus events, whereas in Lowet et al., monkeys made a saccade to the attended stimulus. Because human and monkey subjects tend to make microsaccades in the same direction as their upcoming saccadic choices (*Yu et al., 2016*), the directions of microsaccades in the previous attention task might be related to the motor preparation of the upcoming saccade choice as well as related to the allocation of attention. By using a joystick release, we minimized these lateralized effects related to saccade preparation. These same three factors also help explain why the distribution of microsaccade directions in our study was not strongly biased toward the cue location as found in some previous studies (*Engbert and Kliegl, 2003*; *Hafed and Clark, 2002*; *Laubrock et al., 2010*; *Pastukhov and Braun, 2010*).

How are our findings related to the previous studies showing that attention task performance covaries with microsaccades? First, the main finding from our study is that the attention-related modulation of SC neurons does not require the occurrence of microsaccades. This aspect of our results is entirely consistent with recent work showing that attentional effects are present even in the absence of microsaccades (*Liu et al., 2021*; *Poletti et al., 2017*). Second, we did find that the amplitude of attention-related modulation varied with microsaccade direction when they did occur. Crucially these differences in modulation across microsaccade conditions preceded microsaccade onset. Moreover, the attention-related modulation that preceded microsaccades toward the cued location was commensurate to that found in the no-microsaccade condition. This aspect of our results supports viewing microsaccades as a 'marker' for the state of attention where microsaccade direction may be correlated with the spatial allocation of attention, but is not causally necessary (*Hafed, 2013*). Since some of the same brain structures that control spatial attention are also involved in generating microsaccades (*Hafed et al., 2009*; *Lovejoy and Krauzlis, 2010*), the occurrence and direction of microsaccades would be expected to covary with the allocation of attention as measured in behavioral tasks (*Engbert and Kliegl, 2003*; *Hafed and Clark, 2002*; *Yuval-Greenberg et al., 2014*).

Finally, we acknowledge that tasks like those used in this study are somewhat unnatural. Outside these types of test situations, it would be rare for monkeys to selectively attend a stimulus in the periphery for extended periods of time while maintaining central fixation. Would similar results hold under more natural conditions when subjects look directly at the peripheral stimulus? Of course, without imposing central fixation, the period of peripheral attention would be much shorter and include few if any microsaccades, but we would expect to see similar attention-related neuronal modulation at the peripheral site, consistent with the effects of pre-saccadic modulation of attention (*Li et al., 2021*). Once the attended stimulus is foveated, similar attention-related effects might take place for neurons representing the central visual field, based on recent behavioral studies demonstrating that attention can be selectively distributed even within the fovea (*Poletti et al., 2017*). Considering the now substantial evidence that the foveal portion of the SC map is activated when the behaviorally relevant location is at the center of the visual field (e.g., during parafoveal smooth pursuit as in *Hafed and Krauzlis, 2008*), we expect that SC neurons with foveal RFs would display similar attention-related modulation as we found here. However, further studies

are needed to understand the neuronal mechanisms that control how attention is allocated in and around the fovea.

## Materials and methods

### General

We collected and analyzed data from two adult male rhesus monkeys (*Macaca mulatta*) weighing 9–12 kg. Data collection and analysis were not performed blind to the conditions of the experiments. All experimental protocols were approved by the National Eye Institute Animal Care and Use Committee, and all procedures were performed in accordance with the United States Public Health Service policy on the humane care and use of laboratory animals.

### Behavioral task

Monkey subjects performed a covert spatial attention task which has been described in detail previously (*Herman et al., 2018*; *Herman and Krauzlis, 2017*) and presented in *Figure 1a*. In brief, monkeys initiated each trial by pressing down on a joystick, which triggered the presentation of a central fixation square. After monkeys acquired fixation, a white cue ring flashed (133 ms) in the periphery indicating the 'cued' location. Then 500ms after the cue ring, two color patches were presented on the screen, with one patch in the same location as the previous cue ring (the 'cued patch') and another at equally eccentric location in the opposite visual hemifield (the 'foil patch'). Within each trial, the mean saturation of one of the color patches could possibly change 1–4 s after color patches onset; if the change occurred at the 'cued patch,' the monkeys were required to respond within 150–750 ms by releasing the joystick, and if the change occurred at the 'foil patch' or if no change occurred, the monkeys needed to continue to hold the joystick down. In each block, cue-change, foil-change trials followed a ratio of 3:1 and all the trials were presented in pseudorandom order. The cue conditions were block designed (70 trials in each cued block) and the cued location alternated from block to block. The transition of cued block was indicated by single-patch trials (n=18) with only one color patch inside the 'cued' location. These single-patch trials were also used for later control analyses (*Figure 4—figure supplement 3*).

### SC recordings

We recorded SC extracellular activity in both monkeys using 24-channel or 32-channel V-Probes (50 µm spacing between contacts; Plexon Inc, Dallas, TX). In three sessions in monkey 1, we recorded both the left and right SC simultaneously. In 10 sessions in monkey 2, we used a single V-Probe to record in either the left or right SC. In each session, after advancing the V-Probe into the intermediate/deep layers of SC, we first monitored putative neuronal activity on each recording channel using a threshold-crossing method ($\mu\pm3\sigma$ on each channel). Based on the threshold crossing activity during a visually guided saccade task, we estimated the RFs for the neuronal activity on each recording channel. We then manually set the location of attention-task stimuli to be overlap with estimated RF centers. During the single V-Probe recording sessions in monkey 2, the two color patches were always 180° of elevation apart, with one patch inside the RFs. During the dual V-Probe recording sessions in monkey 1, the two color patches were placed at 0° and 200° of elevation to align stimulus location with both RFs.

Continuous electrophysiological data were acquired (40 kHz sample-rate) and high-pass filtered through an 'Omniplex D' system (Plexon Inc, Dallas, TX). Single units were sorted offline with Kilosort2 (*Pachitariu et al., 2016*). Because our intention was to test the relationship between SC attention-related modulation and microsaccade generation, we focused on SC neurons displaying classic visual attention-related modulation in our covert visual attention task which has been well established previously (*Herman et al., 2018*; *Herman and Krauzlis, 2017*). Neurons were identified as visually responsive if they displayed significant visual-evoked activity (50–150 ms) after color patches onset compared to baseline (−100 to 0 ms before color patches onset) in both single-patch and two-patch trials (p<0.01, Wilcoxon rank-sum test, two-sided). Visually responsive neurons with significant attention-related modulation during the later delay period (200–1000 ms) were included for further analyses (mean firing rate for cue-in-RF was significantly higher than for cue-out-of-RF, p<0.01,

Wilcoxon rank-sum test, two-sided). In total, our data set included 34 units in monkey 1 and 34 units in monkey 2.

## Microsaccade detection

Right eye position was monitored by an EyeLink 1000 infrared eye-tracking system (SR Research, Ottawa, ON, Canada) at a sample rate of 1000 Hz. Microsaccades were initially detected by using the 2D-velocity-based algorithm (relative velocity threshold = 4 and minimum saccade duration = 6 ms) developed by *Engbert and Kliegl, 2003*. Every detected microsaccade was then inspected and visually verified by the experimenter. Microsaccade direction was calculated relative to the cued location (aligned in each session to 0°) in each trial. Microsaccades with directions ±90° (window size 180°) relative to the cued location were grouped as 'microsaccades toward the cued location,' and the other half of microsaccades were grouped as 'microsaccades away from the cued location'.

## Firing rate analysis

Spike counts were binned in non-overlapping 20-ms windows. For normalization, each neuron's spike counts was z-scored (mean subtracted and divided by standard deviation calculated from the neuron's binned counts across trials and conditions).

## Timing match of microsaccades

When determining the attention-related modulation aligned to the onset of individual microsaccades (*Figure 4*), we matched the timing of microsaccades and generated 'no microsaccade' control data. To do so, we first separated our analysis time period (200–1000 ms after color patches onset) into eight non-overlapping 100-ms bins, and computed the temporal distribution of trial counts for each microsaccade and attention condition. More specifically, for each bin, we counted how many trials had microsaccades toward or away from the cued location, and how many trials had no microsaccades ±200 ms relative to the center of this bin. After counting all the trials, we used the lowest trial number within each temporal bin across conditions to subsample the data (without replacement) for all conditions, thereby matching the bin counts in the distributions across all conditions. For the 'no microsaccade' condition, the center of each bin was used to match the timing of microsaccades.

## Statistical analyses

To quantify how the occurrence or absence of microsaccades affected the time course of SC attention-related modulation (*Figure 3*), we took the average normalized firing rates in five different time epochs during the delay period across all the conditions from each neuron (n=68), and performed an ANOVA (total d.o.f.=2039; error d.o.f.=2010) with three factors: (1) time epoch (five different time epochs, time epoch d.o.f.=4), (2) attention conditions (cue-in-RF and cue-out-of-RF, attention conditions d.o.f.=1), and (3) microsaccade conditions (no microsaccades, microsaccades toward the cued location, and microsaccades away from the cued location, microsaccade conditions d.o.f.=2). We used Tukey-Kramer post hoc comparison to test whether there was significant attention-related modulation (mean firing rate for cue-in-RF was significantly higher than with cue-out-of-RF, p<0.05) in each time epoch and each microsaccade-related condition.

To test the significance of attention-related modulation and firing rate suppression around the time of microsaccades (*Figure 4b*), we computed the average normalized firing rate before and after microsaccade onset across conditions in each neuron (n=68), and performed an ANOVA (total d.o.f.=815; error d.o.f.=804) with three factors: (1) time (before and after microsaccade onset, time d.o.f.=1), (2) attention conditions (cue-in-RF and cue-out-of-RF, attention conditions d.o.f=1), and (3) microsaccade conditions (no microsaccades, microsaccades toward the cued location, and microsaccades away from the cued location, microsaccade conditions d.o.f.=2). The Tukey-Kramer post hoc comparison was used to test whether there was significant attention-related modulation (mean firing rate for cue-in-RF was significantly higher than for cue-out-of-RF, p<0.05) and firing rates suppression (mean firing rate after microsaccade was significantly lower than before microsaccade, p<0.05).

To test whether the attention-related modulation was significantly different before and after microsaccades (*Figure 4c*), we calculated the difference in normalized firing rates between cue-in-RF and cue-out-of-RF conditions during both 'before' and 'after' windows in each neuron (n=68), and performed an ANOVA (total d.o.f.=407; error d.o.f.=402) with two factors: (1) time (before and after

microsaccade onset, time d.o.f.=1) and (2) microsaccade conditions (no microsaccades, microsaccades toward the cued location, and microsaccades away from the cued location, microsaccade conditions d.o.f.=2). Tukey-Kramer post hoc comparisons were used to test for significance ($p<0.05$).

To test whether microsaccade-related neuronal suppression after microsaccades depended on cueing condition (*Figure 4d*), we calculated the difference in normalized firing rates between the 'after' and 'before' windows for both cue-in-RF and cue-out-of-RF conditions in each neuron (n=68), and performed an ANOVA (total d.o.f.=407; error d.o.f.=402) with two factors: (1) attention conditions (cue-in-RF and cue-out-of-RF, attention conditions d.o.f.=1) and (2) microsaccade conditions (no microsaccades, microsaccades toward the cued location, and microsaccades away from the cued location, microsaccade conditions d.o.f.=2). Tukey-Kramer post hoc comparisons were again used to test for significance ($p<0.05$).

To address whether peri-microsaccadic attention-related modulation was related to motor effects associated with microsaccade generation (*Figure 4—figure supplement 3*), we performed control analyses using a subset of our data in which only a single patch was presented in the ipsilateral visual field, so that there was no visual stimulus inside the neurons' RFs. We then aligned SC neuronal activity to the onset of microsaccades directed toward and away from the RF of the SC neurons. For comparison, we also analyzed firing rates from timing-matched epochs with no microsaccades. To test whether there were significant changes of activity around the onset of microsaccade, we took the average normalized firing in the 'before' and 'after' windows for each neuron (n=68) and performed an ANOVA (total d.o.f.=407; error d.o.f.=402) with two factors: (1) time (before and after microsaccade onset, time d.o.f.=1) and (2) microsaccade conditions (no microsaccades, microsaccades toward the RF, and microsaccades away from the RF, microsaccade conditions d.o.f.=2). Tukey-Kramer post hoc comparisons were used to test for significance ($p<0.05$).

To exclude the possibility that the peri-microsaccade attention-related modulation was related to the differences in eye position before microsaccades, we matched the eye positions distributions before microsaccades toward and away from the cued location across cue conditions (*Figure 4—figure supplement 3*) and re-quantified the peri-microsaccade attention-related modulation. To match the eye positions, we first calculated the 2D distribution of average eye position in the 'before' window using spatial bins of 0.25°×0.25° across all conditions. We then used a subsampling procedure, like that described above to match the 2D distributions across conditions. We then reanalyzed the average neuronal activity aligned on individual microsaccades for each neuron (n=68) and performed an ANOVA (total d.o.f.=543; error d.o.f.=536) with three factors: (1) time (before and after microsaccade onset, time d.o.f.=1), (2) attention conditions (cue-in-RF and cue-out-of-RF, attention conditions d.o.f.=1), and (3) microsaccade conditions (microsaccades toward the cued location and microsaccades away from the cued location, microsaccade conditions d.o.f.=2). Tukey-Kramer post hoc comparisons were used to test whether there was significant attention-related modulation ($p<0.05$).

## Acknowledgements

The authors thank Nick Nichols, Daniel Yochelson, Denise Parker, and Amber Lopez for technical support. The authors thank Xuefei Yu, Lupeng Wang, Christian Quaia, Sheridan Goldstein, Kerry McAlonan, and Kara Cover for helpful discussions. This study was supported by the National Eye Institute Intramural Research Program at the National Institutes of Health (ZIA EY000511).

## Additional information

### Funding

| Funder | Grant reference number | Author |
|---|---|---|
| National Eye Institute | ZIA EY000511 | Richard J Krauzlis |
| National Institutes of Health | | Richard J Krauzlis |

The funders had no role in study design, data collection and interpretation, or the decision to submit the work for publication.

### Author contributions

Gongchen Yu, Conceptualization, Data curation, Formal analysis, Software, Visualization, Writing - original draft, Writing – review and editing; James P Herman, Richard J Krauzlis, Conceptualization, Data curation, Formal analysis, Software, Writing – review and editing; Leor N Katz, Data curation, Writing – review and editing

### Author ORCIDs

Gongchen Yu http://orcid.org/0000-0001-9216-8174
James P Herman http://orcid.org/0000-0001-6916-2807
Leor N Katz http://orcid.org/0000-0002-2742-6533
Richard J Krauzlis http://orcid.org/0000-0002-1826-0447

### Ethics

All experimental protocols (#NEI-649) were approved by the National Eye Institute Animal Care and Use Committee, and all procedures were performed in accordance with the United States Public Health Service policy on the humane care and use of laboratory animals.

### Decision letter and Author response

Decision letter https://doi.org/10.7554/eLife.74168.sa1
Author response https://doi.org/10.7554/eLife.74168.sa2

## Additional files

### Supplementary files

• Transparent reporting form

### Data availability

Data for all the figures are available via Dryad http://doi.org/10.5061/dryad.83bk3j9s9.

The following dataset was generated:

| Author(s) | Year | Dataset title | Dataset URL | Database and Identifier |
|---|---|---|---|---|
| Yu G, Herman JP, Katz LN, Krauzlis RJ | 2022 | Data from: Microsaccades as a marker not a cause for attention-related modulation | http://dx.doi.org/10.5061/dryad.83bk3j9s9 | Dryad Digital Repository, 10.5061/dryad.83bk3j9s9 |

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
