## [Editor Report]

This is very much needed work, especially in light of the recent debate regarding whether or not microsaccades are the cause of peripheral attentional effects. A few influential papers have been published recently strongly suggesting that attentional effects are primarily the result of the execution of tiny microsaccades that humans/primates perform during fixation while attending to peripheral stimuli. These past findings have, naturally, a number of implications for the way we interpret visual attention, and raised the question of whether shifts of attention are dependent on microsaccades. By explicitly comparing and quantifying the effects of attention on neuronal responses in the presence and in the absence of microsaccades, this work provides important insights into this debate.

---

## [Decision Letter]

**Decision letter after peer review:**

Thank you for submitting your article "Microsaccades as a marker not a cause for attention-related modulation" for consideration by *eLife*. Your article has been reviewed by 2 peer reviewers, and the evaluation has been overseen by Jennifer Groh as the Reviewing Editor and Joshua Gold as the Senior Editor. The following individuals involved in review of your submission have agreed to reveal their identity: Martina Poletti (Reviewer #1); Martin Rolfs (Reviewer #2).

Essential revisions:

See detailed reviews below: both reviewers concurred about the importance of the work and the high quality of the manuscript. They suggest it could be strengthened further with:

1. Additional analyses verifying the effectiveness of the attention manipulation

2. Additional analyses verifying the effectiveness of the saccade detection

3. Additional commentary related to certain aspects of the findings and/or their relationship to existing literature

4. Further consideration of the relationship between microsaccades and the location of the stimuli and the impact on neural responses and extension to natural, free-viewing*Reviewer #1 (Recommendations for the authors):*

1. I praise the effort made by the authors in explaining possible causes of discrepancies between their and Lowets et al., findings. However, I would have liked the authors to elaborate more on how their findings relate with the current literature showing that microsaccades modulate attentional effects in humans and that attentional effects are minimal or absent in the absence of microsaccades directed to the cued location (e.g., Hafed 2013).

2. Lines 226-229, can these effects be described as the results of pre-microsaccadic attention?

3. To further link this work with the current literature on this topic, these findings are coherent with work showing that attentional effects are present at the behavioral level even in the absence of microsaccades (e.g., Poletti et al., 2017).

4. Supplementary Figure 1. I suggest adding legends in the graphs.

*Reviewer #2 (Recommendations for the authors):*

1. Manipulation check

It would be great to see the behavioral data on detection of the color change. Does the modulation of firing rates depend on the subjects’ response? The authors chose to name it attention-related modulation, but this assumes that the cue was successful in manipulating attention (i.e., a behavioral advantage in sensitivity to changes at the cued location) in the first place. One way to confirm the link of the neural results to attention (rather than just the cue) is to analyze neural firing rates as a function of response rather than cue location (provided subjects made enough errors). Trials in which the monkey responds to the foil should show the opposite pattern of firing rates (higher firing rates for the “Cue out of RF” condition as compared to “Cue in RF”, or at least a reduced modulation in favor of the “Cue in RF” condition).

2. Reliability of microsaccade detection

One of the main results of the study is that attention-related modulations were observed even in the absence of microsaccades. Thus, the authors need to ensure that their results are not a consequence of failing to detect microsaccades. Poletti and Rucci (Vision Res, 2016; https://doi.org/10.1016/j.visres.2015.01.018) discuss a number of ways to address this concern. One way would be to use a more lenient / liberal detection criterion (include lower-velocity events that might not actually constitute a microsaccade) and see if the main results of the study depend on that choice. Another, possibly more direct, way of looking at this is to turn the authors’ analyses around and analyze eye velocity as a function of the recorded neuron’s firing. I am not an expert here, but would it be possible to compute a spike-triggered average of eye velocity in the no-MS trials when considering the first spike after the onset of the color patches in the “In RF” condition and compare it to timing-matched eye velocities in the “Outside of RF” trials? The authors may come up with additional ideas or reasons that render this concern less severe.

3. Relation to previous claims of causality

The authors should discuss their findings with respect to those of Hafed (2013, Neuron).

---

## [Author Response]

Essential revisions:See detailed reviews below: both reviewers concurred about the importance of the work and the high quality of the manuscript. They suggest it could be strengthened further with:1. Additional analyses verifying the effectiveness of the attention manipulation

We now document the effectiveness of the attention manipulation by plotting the hit and false-alarm rates for each subject in each experimental session. We also confirm the link between SC neuronal activity and attention allocation by showing that the size of the attention-related modulation varies with the subjects’ success in the task. We also note that the same attention manipulation has been used in previous studies examining the neuronal mechanisms of attention.

2. Additional analyses verifying the effectiveness of the saccade detection

We have performed additional microsaccade detection analyses using both more stringent and more lenient thresholds (the "λ" value of Engbert and Kliegl, 2003). We have verified that our findings are robust over a range of detection thresholds.

3. Additional commentary related to certain aspects of the findings and/or their relationship to existing literature

We have modified the main text based on the suggestions of the referees, substantially expanding the relevant sections of the Discussion.

4. Further consideration of the relationship between microsaccades and the location of the stimuli and the impact on neural responses and extension to natural, free-viewing

We now devote a new paragraph in the Discussion to this point.

Reviewer #1 (Recommendations for the authors):1. I praise the effort made by the authors in explaining possible causes of discrepancies between their and Lowets et al., findings. However, I would have liked the authors to elaborate more on how their findings relate with the current literature showing that microsaccades modulate attentional effects in humans and that attentional effects are minimal or absent in the absence of microsaccades directed to the cued location (e.g., Hafed 2013).

The reviewer raises an interesting point, although we would like to be careful about overstating the relevance to the human literature on microsaccades and attention.

The main conclusion from our study is that attention-related modulation of neuronal activity does not require the occurrence of microsaccades. This aspect of our results is entirely consistent with recent work showing that attentional effects are present at the behavioral level even in the absence of microsaccades (Poletti et al., 2017).

We also find that the size of the modulation varies with the direction of the microsaccades when they do occur – but crucially these differences in modulation precede the onset of the microsaccade. This part of our results forms the basis of our conclusion that microsaccades are a ‘marker’ for the state of attention but not causally responsible. We believe these findings are entirely compatible with the findings in the human literature about the covariation of microsaccades and attention task performance: since microsaccades are a marker for the state of attention, their occurrence and direction correlate with behavioral performance on attention tasks (Engbert and Kliegl, 2003; Hafed and Clark, 2002; Yuval-Greenberg, Merriam, and Heeger, 2014).

However, we also recognize that the relationship we found between microsaccades and attention-related modulation is still correlational and cannot rule out the possibility that the attention-related modulation around microsaccade generation is specifically related to the generation of microsaccades (Hafed, 2013). We have answered this point in detail in the point #2 related to the pre-microsaccadic attention right below.

We now address these points in the third paragraph of discussion (lines 300-317).

2. Lines 226-229, can these effects be described as the results of pre-microsaccadic attention?

We cannot rule out the possibility that these effects are related to pre-saccadic or pre-microsaccadic attention, although we note that the attention-related modulation preceding microsaccades towards the cued location was about the same as that found in the no microsaccade condition. The similarity in the amplitude of the modulation does not give us a reason to claim the presence of a special pre-microsaccadic effect (Hafed, 2013). Nonetheless, we now address this possibility in the third paragraph of discussion (lines 305-313).

3. To further link this work with the current literature on this topic, these findings are coherent with work showing that attentional effects are present at the behavioral level even in the absence of microsaccades (e.g., Poletti et al., 2017).

Thank you for pointing that out. We now mention and cite the paper (line 305).

4. Supplementary Figure 1. I suggest adding legends in the graphs.

Thanks for the suggestion. We have added the legends.

Reviewer #2 (Recommendations for the authors):1. Manipulation checkIt would be great to see the behavioral data on detection of the color change. Does the modulation of firing rates depend on the subjects' response? The authors chose to name it attention-related modulation, but this assumes that the cue was successful in manipulating attention (i.e., a behavioral advantage in sensitivity to changes at the cued location) in the first place. One way to confirm the link of the neural results to attention (rather than just the cue) is to analyze neural firing rates as a function of response rather than cue location (provided subjects made enough errors). Trials in which the monkey responds to the foil should show the opposite pattern of firing rates (higher firing rates for the "Cue out of RF" condition as compared to "Cue in RF", or at least a reduced modulation in favor of the "Cue in RF" condition).

Many thanks for your helpful suggestions.

To address the reviewer’s comments, we have added a new plot (Figure 1b) showing the hit and false-alarm rate for each subject in each experimental session.

We also now point out (lines 95-96) that the amplitude of the change was adjusted to be just slightly above the threshold for detection; hence, the hit rates were generally between 75-90%. The performance was very consistent across sessions in our well-trained monkeys, and the low rate of false alarms for ‘foil’ changes provides behavioral confirmation that they attended to the correct stimulus location (described on lines 93-99).

We also followed your recommendation and added new Figure 1d showing how SC neuronal attention-related modulation varied with the subjects’ behavioral response. We found that SC neurons displayed higher attention-related modulation when subjects correctly responded to the cued change (hit trials) compared to when subjects missed the cued change (miss trials). This confirms that the attention-related modulation we observed was indeed linked to performance in our attention task (described on lines 110-118).

Finally, we also briefly note in the results (lines 84-86 and lines 117-118) that the same attention manipulation has been used in previous studies examining the neuronal mechanisms of attention, including studies that used causal manipulations that establish the link between SC activity and attention task performance.

2. Reliability of microsaccade detectionOne of the main results of the study is that attention-related modulations were observed even in the absence of microsaccades. Thus, the authors need to ensure that their results are not a consequence of failing to detect microsaccades. Poletti and Rucci (Vision Res, 2016; https://doi.org/10.1016/j.visres.2015.01.018) discuss a number of ways to address this concern. One way would be to use a more lenient / liberal detection criterion (include lower-velocity events that might not actually constitute a microsaccade) and see if the main results of the study depend on that choice. Another, possibly more direct, way of looking at this is to turn the authors' analyses around and analyze eye velocity as a function of the recorded neuron's firing. I am not an expert here, but would it be possible to compute a spike-triggered average of eye velocity in the no-MS trials when considering the first spike after the onset of the color patches in the "In RF" condition and compare it to timing-matched eye velocities in the "Outside of RF" trials? The authors may come up with additional ideas or reasons that render this concern less severe.

We agree it is important to verify the reliability of our microsaccade detection and thank you for the helpful recommendations.

We have followed your suggestion and added supplementary Figure 3 to address this point. We repeated the microsaccade detection analyses using 5 different thresholds (the "λ" value of Engbert and Kliegl, 2003). The results from these additional analyses verify the robustness of the findings we report – our conclusions hold across the range of more lenient and more strict detection thresholds (described on lines 217-218).

3. Relation to previous claims of causalityThe authors should discuss their findings with respect to those of Hafed (2013, Neuron)

We agree and this point was also made by Reviewer #1.

Hafed 2013 presented evidence that microsaccades might be responsible for the performance changes in humans commonly attributed to covert shift of attention.

Our study addresses a related issue about the relationship between microsaccades and attention-related neuronal modulation in the monkey SC and our results suggest that microsaccades can be used as a marker for the state of attention but are not a prerequisite.

We recognize that the relationship we found between microsaccades and attention-related modulation is still correlational, and that the generation of microsaccades might be important for the presence of attention-related modulation around microsaccade. However, the attention-related modulation preceding microsaccades towards the cued location was about the same as that found in the no microsaccade condition. The similarity in the amplitude of the modulation does not give us a reason to claim the presence of a special pre-microsaccadic effect (Hafed, 2013).

As mentioned above, we now address these points in the third paragraph of discussion (lines 300-317).